# Functional PET Neuroimaging in Consciousness Evaluation: Study Protocol

**DOI:** 10.3390/diagnostics13122026

**Published:** 2023-06-10

**Authors:** Tom Paunet, Denis Mariano-Goulart, Jeremy Deverdun, Emmanuelle Le Bars, Marjolaine Fourcade, Florentin Kucharczak

**Affiliations:** 1Department of Nuclear Medicine, Gui de Chauliac Hospital, Montpellier University Hospital Center, University of Montpellier, 34090 Montpellier, France; 2I2FH, Department of Neuroradiology, Gui de Chauliac Hospital, Montpellier University Hospital Center, University of Montpellier, 34090 Montpellier, France

**Keywords:** ^18^F-FDG, PET, constant infusion, glucose metabolism, stimuli, functional PET, fPET, dynamic PET, disorder of consciousness, coma

## Abstract

Ensuring a robust and reliable evaluation of coma deepness and prognostication of neurological outcome is challenging. We propose to develop PET neuroimaging as a new diagnostic and prognosis tool for comatose patients using a recently published methodology to perform functional PET (fPET). This exam permits the quantification of task-specific changes in neuronal metabolism in a single session. The aim of this protocol is to determine whether task-specific changes in glucose metabolism during the acute phase of coma are able to predict recovery at 18 months. Participation will be proposed for all patients coming for a standard PET-CT in our center in order to evaluate global cerebral metabolism during the comatose state. Legally appointed representative consent will be obtained to slightly modify the exam protocol: (1) ^18^F-fluorodeoxyglucose (^18^F-FDG) bolus plus continuous infusion instead of a simple bolus and (2) more time under camera to perform dynamic acquisition. Participants will undergo a 55-min fPET session with a 20% bolus + 80% infusion protocol. Two occurrences of three block (5-min rest, 10-min auditory stimulation and 10-min emotional auditory stimulation) will be performed after reaching equilibrium of FDG arterial concentration. We will compare the regional brain metabolism at rest and during the sessions of auditory and emotional auditory stimulation to search for a determinant of coma recovery (18 months of follow-up after the exam). Emotional auditory stimulation should induce an activation of: the auditory cortex, the consciousness areas and the neural circuitry for emotion (function to coma deepness). An activation analysis will be carried out to highlight regional brain activation using dedicated custom-made software based on Python statistical and image processing toolboxes. The association between activation levels and the Coma Recovery Scale-Revisited (CRS-R) will be assessed using multivariate analysis. If successful, the results from this study will help improve coma prognosis evaluation based on the pattern of neuronal metabolism at the onset of the pathology. The study protocol, rationale and methods are described in this paper.

## 1. Introduction

### 1.1. Coma and Prognosis Evaluation

Coma prognosis evaluation is of great importance for the patient’s future management (rehabilitation, life-support care) and life or death decisions. To establish a prognosis for the disorders of consciousness, they are categorized into several states, depending on arousal (vigilance) and awareness (perception of the environment and of the self) [1,2]. Classic nosology from coma (lack of spontaneous opening of the eyes and lack of wakefulness) to recovery begins with the vegetative state (VS) [3,4], which is also called unresponsive wakefulness [5]: patients are awake but are unaware of internal or external stimuli. The next state is the minimally conscious state [6]: patients show signs of arousal with a certain degree of awareness with basic communication possible in the minimally conscious state plus (MCS+) [7], which is contrary to the minimally conscious state minus (MCS-) [8]. Consciousness recovery may occur after months or years [9]. Two other diagnostic entities might be cited: brain death [10], with irreversible brainstem destruction and no arousal or awareness, and locked-in syndrome [11]: with quadriplegia and only eye-coded communication.

These nosological distinctions are challenging. Indeed, bedside clinical evaluation is the gold standard to assess the consciousness level, but it frequently leads to misdiagnosis [12,13]. For example, distinguishing MCS- from VS could be particularly complex. Moreover, not every patient experiences all the transitions between coma, unresponsive wakefulness syndrome, MCS- and MCS+ [8]. Thus, at least 13 behavioral assessment scales for disorders of consciousness have been proposed for clinical evaluation [14].

The recently developed neuroimaging techniques have brought new insights on brain function after a coma and can complement clinical assessment by improving the diagnosis and prognosis of patients with disorders of consciousness [15,16].

### 1.2. Functional PET

Standard cerebral PET shows the brain metabolism representing all the neuronal processes during the acquisition period following a bolus injection of ^18^FDG. No temporal information is available: to determine changes in ^18^FDG uptake after a stimulus, two PET exams are needed (at baseline and one after stimulus). In addition to the supplementary radiation exposure, numerous factors can interfere in the result (mis-registration, changes in basal state due to uncontrolled variables during the two PET sessions). ^18^FDG-PET can be used in a more dynamic manner and is capable of detecting changes in glucose metabolism within a single imaging session. Villien et al. named this technique functional PET (fPET) [17].

PET imaging was historically the first technique to demonstrate neuronal activation measuring regional cerebral blood flow (rCBF) with oxygen-15 labeled water [18]. This was supplanted by functional magnetic resonance imaging (fMRI) for several reasons: no exposure to ionizing radiation and superior spatial and especially temporal resolution, which allows for the dynamic measurement of multiple responses within a single imaging session [19].

However, interest in fPET has been growing in the past several years. Several authors have proposed injection/acquisition protocols to highlight regional neuronal activation during a single PET session with continuous infusion [17] or bolus plus continuous infusion [20]. These protocols with recent PET scanners make single session study possible with improved sensitivity and sensitivity for studying brain connectivity [21,22]. One of the challenges for the more widespread use of fPET is the need for a safe and reliable system for ^18^FDG administration (bolus plus infusion).

The aim of our protocol is to determine whether the neuronal metabolism implicated in the consciousness circuit can predict recovery in patients with recent coma. We will investigate the activation of the whole brain during an emotional auditory stimulation.

## 2. Materials and Methods

### 2.1. Study Design

This monocentric prospective study is expected to recruit 25 subjects over 24 months. As it does not entail a potential change in patient care, the protocol is considered a non-interventional study, and it has been validated by the French Nuclear Medicine Research Ethics Committee (CEMEN 2022-02). The study will be conducted in accordance with the Declaration of Helsinki. Given that it is focused on disorders of consciousness, informed consent will be obtained from legally appointed representatives.

#### 2.1.1. Population

The protocol will be proposed to all patients without exclusion criteria and referred to our nuclear medicine department for a cerebral ^18^FDG PET to evaluate global brain-metabolism in the acute phase of coma.

#### 2.1.2. Inclusion Criteria

We will consider for inclusion patients over 18 years old with a recent disorder of consciousness (less than 30 days). A clinical assessment of the JFK Coma Recovery Scale–Revised [23] (CRS-R) from a senior neurologist (or resuscitation physician) and a volunteer for sensory stimulation (see below) are needed.

#### 2.1.3. Exclusion Criteria

Exclusion criteria include: pregnancy, extensive cerebral lesion which prevents recognition of cerebral functional areas, and inability to fulfill a quality imaging criterion (movements during PET acquisition, for example).

#### 2.1.4. Conduct of the Study

The study protocol will be proposed to all patients’ legally appointed representatives when we receive a PET-CT request to assess global brain metabolism during coma. Following signatures of informed consent, subjects meeting all inclusion criteria and no exclusion criteria will undergo a PET-CT session under a bolus plus infusion protocol. This protocol allows for a final standard interpretation (summed images) and produces temporal data we will use to functionally interpret the exam. Each legal representative will be asked to participate in the emotional auditory stimulation during the exam. Clinical assessment of CRS-R will be retrieved from medical records at 6, 12 and 18 months.

### 2.2. Experiment Protocol

#### 2.2.1. Bolus Plus Infusion Paradigm

We will use a bolus plus continuous infusion protocol for fPET acquisition that has demonstrated its ability to assess neuronal activation on small human samples [17,20]. This type of administration eliminates the need for arterial FDG concentration monitoring [22]. To ensure that the constant activity in the carotids hypothesis is verified in each exam, we will statistically verify the stationarity of the carotid TAC with the augmented Dickey–Fuller test [24]. The protocol is summarized in Table 1. Task blocks (rest, listening to an unknown language, emotional auditory simulation) begin 5 min after infusion start to reach a baseline state for vascular activity [22].

#### 2.2.2. Remote Syringe Pump Prototype

To control the injection sequence, we developed a prototype based on a commercial electronic syringe pump (Agilia Injectomat, Fresenius Kabi (Bengaluru, India)). We therefore developed a remotely controlled system to safely operate it. The system was built around a Raspberry Pi 3 Model B+ (Raspberry Pi Foundation (Cambridge, UK)). This small and affordable single-board computer was originally designed to promote the teaching of basic computer science. However, its low cost, modularity, and open design have prompted its wide use in robotics, for example. Whole inputs (filling sensor, camera) and outputs (servomotors, relay board) are connected to its dedicated interface: a general-purpose input–output (GPIO) connector. Its operating system is based on Linux, and it can be controlled with a mouse and a keyboard via a remote display. In our case, the infusion system (automated injector, modified syringe pump, Raspberry-Pi) is installed next to the PET camera. The control screen and input peripherals are connected to the Raspberry-pi and installed in the PET operator room (radiation-free).

A Python3 script with graphical user interface was written to operate inputs and outputs. The syringe pump screen is captured and sent to the control system screen.

Figure 1 shows the infusion line from the infusion pump to the patient; all the preparation (connection, purge…) can be performed manually before radio tracer arrival in the room.

### 2.3. Stimulation during PET Exam

We chose to use an auditory stimulation with an affective connotation to expose the induced metabolism in acoustic and consciousness areas. Regional blood flow changes were demonstrated in relation to affective speech presentation in case of persistent VS [25]. A simple auditive stimulation (domestic storytelling) during fMRI induced specific brain activity which could help with discerning a partial awareness during VS (as occurs in MCS) [26]. Moreover, this simple stimulation is easy to set up for comatose patients with frequently closed eyes. Each stimulation session lasts 20 min (10 min for auditory and 10 min for emotional auditory stimulations), which is preceded by 5 min of rest. This 25-min block will be repeated two times at 5 and 30 min (Figure 2). We will ask a family member of the patient to tell him a personal story with affective connotation during these two stimulation blocks. The family member will speak to the patient through the PET speakers, from the PET operator’s room, so that she/he will not be exposed to ionizing radiation.

### 2.4. Follow-Up

Clinical assessment of CRS-R will be retrieved from medical records at 6, 12 and 18 months.

### 2.5. PET Data Acquisition

All PET scans will be performed with a Siemens Biograph mCT20 flow (Siemens AG, Munich, Germany). The patient’s medical team will be instructed to stop enteral or parenteral nutrition at least 6 h before scanning. Patients will receive an i.v. line for tracer application. Blood glucose will be measured before scanning with a glucometer before patient installation in the PET room. PET data will be acquired in list mode.

Acquisition begins at t = 0 s. To improve the signal-to-noise ratio for early time frames, we used the administration of a 20% bolus prior to constant infusion [22]. The syringe pump prototype is programmed to perform a 20% bolus of the ^18^FDG total activity at t = 10 s, which is followed by continuous infusion until the end of the acquisition.

List-mode data will be reconstructed with an ordinary Poisson-ordered subset expectation maximization algorithm (OP-OSEM, 3 iterations, 21 subsets) with a frame duration of 30 s (note that this setup can be modified later to optimize the sensitivity of the PET analysis). The attenuation correction maps will be obtained from the low-dose CT scan.

The data will be reconstructed with an isotropic voxel size of 2 mm into a volume consisting of 109 transverse slices of 400 × 400 pixels. All volumes will be smoothed using a 3D filter with a 10 mm FWHM isotropic Gaussian kernel.

### 2.6. fPET Paradigm

To asses cortical activation, we measure the difference in CMRGlu during rest and stimulation blocks (auditory or emotional). In this section, we describe the kinetic analysis used and the goal of bolus + continuous infusion.

#### 2.6.1. Kinetics Analysis for Classic Bolus Method

The usual two-tissue compartmental model with irreversible uptake is used to describe the signal from cerebral ROI (Figure 3). A simplified Patlak method [27,28] is used to resolve differential equations which describe the system. This method can be applied if some simplifications are made: ^18^FDG phosphorylation is irreversible ($k_{4}=0$), vascular fraction is neglected (f=0), and the reversible compartment is in steady-state equilibrium ($\frac{dC_{nm}\left(t\right)}{dt}=0$).

The first two proposals ($k_{4}=0$ [29] and f=0 [27]) are commonly accepted for cerebral tissue.

For the hypothesis $\frac{dC_{nm}\left(t\right)}{dt}=0$, the issue is more complicated. This equilibrium could not be verified especially during stimulation–rest tests which induce a break in the steady state. This assumption is systematically made in pharmacokinetic models combined with Michaelis–Menten kinetics applied to the first phosphorylation by hexokinase [30,31]. This hypothesis allows us to deduce CMRGlu in the form given in Equation (Equation 7).

The Patlak plot requires that the ^18^FDG in the reversible compartments is in steady state with the ^18^FDG in the blood plasma [28].

#### 2.6.2. Kinetics Analysis for Infusion + Bolus Method

Villien et al. [17] suggested that during the task, a new steady state will be reached after a transient disruption of the equilibrium concentrations during the initiation of a task. The mathematical implications are explained below.

Using the two-tissue compartmental model, the following differential equations describe the system [32]:(1)$$\frac{{dC}_{ROI}\left(t\right)}{dt}=\;\frac{{dC}_{nm}\left(t\right)}{dt}+\;\frac{{dC}_{m}\left(t\right)}{dt}$$
(2)$$\frac{dC_{nm}\left(t\right)}{dt}=K_{1}C_{v}\left(t\right)-\left(k_{2}+k_{3}\right){\cdot C}_{nm}\left(t\right)$$
(3)$$\frac{dC_{m}\left(t\right)}{dt}=k_{3}C_{nm}\left(t\right)\;$$
with $C_{ROI}$ being the ROI ^18^FDG concentration, $C_{nm}$ being the non-metabolized ^18^FDG concentration in the reversible compartment, $C_{m}$ being the metabolized ^18^FDG concentration in the irreversible compartment, and $C_{v}$ being the ^18^FDG concentration in plasma.

During a stimulation or a rest interval, there are different steady states, which means that $\frac{dC_{nm}\left(t\right)}{dt}=0$. In Equation (Equation 2), under these conditions:(4)$$\frac{dC_{nm}\left(t\right)}{dt}=K_{1}C_{v}\left(t\right)-\left(k_{2}+k_{3}\right){\cdot C}_{nm}\left(t\right)=0$$

Since we have made the assumption that $C_{nm}$ is a constant (outside the transition phases), that implies that $C_{v}$ must also be a constant due to the ^18^FDG infusion + bolus protocol. Given Equation (Equation 4), Equation (Equation 3) yields to:(5)$$\frac{dC_{m}}{dt}\left(t\right)=\frac{K_{1}\cdot k_{3}}{k_{2}+k_{3}}C_{v}\left(t\right)$$

Furthermore, as $\frac{dC_{nm}\left(t\right)}{dt}=0$ in Equation (Equation 1), then Equation (Equation 5) becomes:(6)$$\frac{dC_{ROI}}{dt}\left(t\right)=\frac{dC_{m}\left(t\right)}{dt}=\frac{K_{1}\cdot k_{3}}{k_{2}+k_{3}}C_{v}\left(t\right)$$

By introducing the *CMR_Glu_* formula, it gives:(7)$$\frac{dC_{ROI}}{dt}\left(t\right)=\frac{K_{1}\cdot k_{3}}{k_{2}+k_{3}}C_{v}\left(t\right)={CMR}_{glu}\cdot \frac{LC}{Gly}C_{v}\;\;\mathrm{with}\;{CMR}_{glu}=\frac{K_{1}\cdot k_{3}}{k_{2}+k_{3}}\cdot \frac{Gly}{LC}$$
with $C_{v}$ being the constant value of $C_{v}\left(t\right)$ in the considered phase, LC being the Lumped constant, and Gly being the patient’s glucose level also constant in the considered phase.

Finally, the TAC slope’s in a given ROI is proportional to the *CMR_Glu_*.

#### 2.6.3. Activation Analysis in fPET

Cortical activation corresponds to the variation of *CMR_Glu_* between the resting state (rest) and the stimulated state (stim) of a given brain ROI/voxel:(8)$$\Delta \%CMR_{Glu}=\frac{{\mathrm{CMR}}_{Glu}^{stim}-{\mathrm{CMR}}_{Glu}^{rest}}{{\mathrm{CMR}}_{Glu}^{rest}}\times 100\%$$

Assuming that $C_{v}$ is constant between resting and stimulation periods, this can be approximated from the change in TAC slopes (Equation (Equation 8)):(9)$$\Delta \%CMR_{Glu}=\left(\frac{\frac{dC_{ROI}^{stim}}{dt}}{\frac{dC_{ROI}^{rest}}{dt}}-1\right)\times 100\%$$

### 2.7. Data Analysis

All proposed processing steps will be carried out using custom-made processing software developed with Python 3.9.13.

#### 2.7.1. Pre-Processing

Firstly, since the acquisition may be subject to motion due to its long duration, the 4D volumes are realigned with each other using a rigid transformation. In order to have anatomical correspondences for the activated ROI/voxels in fPET, it is necessary to spatially normalize the 3D + t PET data into the MNI space [33]. To achieve this, the last 60 temporal series (which corresponds to a 30 min static acquisition 25 min after injection) are extracted and summed into a 3D volume, which will be realigned to the SPM PET template (91 × 109 × 91 voxels) [34]. The transformation that minimizes the mutual information between the two volumes is then applied to all the 3D volumes corresponding to the temporal series.

#### 2.7.2. TAC Extraction

Once all volumes are in the same space, the data will be processed in two different ways: the first one involves extracting a TAC for each anatomical ROI from the AAL atlas [35]. The considered TAC is the average of the TACs of all voxels within the ROI. The second method consists of processing all the TACs of all the voxels and is more suitable for visual analysis.

#### 2.7.3. TAC Slopes’ Comparison–First Order Analysis

In the fPET literature, information on the activation of a voxel or region for a specific task is extracted by modeling the problem following the fMRI paradigm and would be written, for our study, as follows: (10)$$TAC\left(t\right)=\beta_{base}^{}X_{base}{\left(t\right)}^{}+\beta_{aud}^{}X_{aud}^{}\left(t\right)+\beta_{emo}^{}X_{emo}^{}\left(t\right)+\beta_{noise}^{}X_{noise}^{}\left(t\right)+\varepsilon ,$$
where $\beta_{}^{}$ represents scalar coefficients, $\beta_{base}^{}X_{base}^{}\left(t\right)$ represents the baseline metabolic signal [17] when no stimulation is conducted, $\beta_{aud}^{}X_{aud}^{}\left(t\right)$ represents the auditory induced activation signal (listening to a word list in an unknown language), $\beta_{emo}^{}X_{emo}^{}\left(t\right)$ represents the auditory + emotional induced activation signal (listening to a close relative recounting a story with affective speech), $\beta_{noise}^{}X_{noise}^{}\left(t\right)$ represents a nuisance regressor (head motion, physiological noise…) and ε represents a noise term not explicated by the model. In this analysis, all TACs are subtracted by a tendency curve (from native data recorded during resting periods) which enables switching from a PET signal accumulating during time to an on–off task design matrix similar to fMRI.

In this study, however, we prefer to use an fPET dedicated analysis that does not require subtracting a trend curve, which can be questionable in its definition. The activation problem of a voxel or ROI for a specific task could be summarized as the problem of regression slopes comparison. Numerous approaches exist in statistics, and we have decided to implement two of them to verify the sensitivity of each in our application.

The first approach is based on the calculation of a score that we called “*s-score*” for “slope score” by comparing the slopes estimated by linear regression corresponding to the stimulation stim={aud,emo} and the baseline stimulation base as: (11)$$s\text{-}\mathrm{score}=\frac{{\mathrm{slope}}_{stim}-{\mathrm{slope}}_{ref}}{\sqrt{sd_{stim}^{2}+sd_{ref}^{2}}},$$
with *sd* corresponding to the standard error of the prediction. Thus, one value of *s-score* per voxel/ROI per stimulation type (auditory or emotional auditory) is estimated. As the stimulation blocks are repeated three times in the proposed protocol, an intermediary step is needed: an initial regression is performed on all blocks of the same stimulation type to ensure that the regression of each set of points has a zero intercept, allowing in a second step to apply Formula (Equation 11).

The second approach is based on the calculation of so called p-score which is derived from a generalized linear model (GLM) that computes whether the stimulation state variable (to each frame corresponds one state of stimulation among “*base*”, “*aud*” and “*emo*”) is an interaction factor between TAC and time variables, otherwise known as *Potthoff analysis* [36]. As for the *s-score*, one value of *p*-score (and associated *p*-value) per voxel/ROI per stimulation type (auditory or auditory + emotional) is estimated.

The estimated activation scores are then thresholded as a function of the desired sensitivity for the visual analysis. The glass-brain and 3D visual of activation maps are generated using the Nilearn Python library [37].

#### 2.7.4. Second-Order Analysis

Association between the activation levels and CRS-R will be performed using multivariate regression adjusted for coma duration and type, age and sex. All voxel-wise statistics will be corrected for multiple comparisons at *p* < 0.001.

#### 2.7.5. Additional Analysis: Benefits of Repeated Stimulation Occurrences

Despite the fact that fTEP technology is recent and still at the research stage, some studies have focused on optimizing certain parameters intrinsic to the technique, such as the minimum stimulation time required [22], different types of stimulation (visual and motor) [38], and the most appropriate type of injection (bolus, infusion, bolus + infusion) [17,22]. However, to the best of our knowledge, the paradigm of repeating rest and stimulation blocks has never been challenged. Initially proposed by researchers coming from the functional MRI research field, the fTEP paradigm was naturally modeled on that of fMRI by performing occurrence repetitions (necessarily longer and fewer in number due to the nature of PET and the minimum frame duration required for an acceptable signal-to-noise ratio).

Thus, as a complementary analysis, we would like to compare activation maps when only one stimulation occurrence is used for statistical analysis instead of two. In order to remain within ethically reasonable acquisition times for our patients, we have decided to limit ourselves to two stimulation occurrences, with longer stimulation times for the activation blocks, so as to check whether, with this particular population and this type of stimulation, the 2-minute duration proposed by [22] is sufficient or whether a longer duration makes the analysis more sensitive. Our hypothesis is that, indeed, a longer duration will be necessary: that is why we propose extending the duration of active stimulation blocks to 10 min.

## 3. Expected Results

To date, fPET has only been performed on healthy subjects to highlight basic sensory or motor cortex activation. Visual stimulation was assessed with an infusion protocol [17,38] and with a bolus + infusion protocol [20,22]. Motor-related areas were tested under motor stimulation [38].

In this study, we will apply the fPET technique to comatose patients in order to highlight a more complicated cortical activation during auditory and auditory + emotional auditory stimulation. A bolus (20%) + infusion (80%) protocol will be used without arterial blood sampling [22], and the expected temporal resolution will allow for a 10-minute task duration [22]. Two occurrences of blocks (no stimulation for 5 min, auditory stimulation for 10 min, and emotional auditory stimulation for 10 min) will be made to allow for an fMRI-like GLM analysis. Cortical maps of the glucose metabolism response to external auditory stimuli will be analyzed with the CRS-R value at 6, 12 and 18 months in order to look for a metabolic pattern linked to the CRS-R evolution. The CRS-R scale was chosen because it is a numeric continuous scale widely used in acute and late coma stages. A GLM and statistical analysis will be applied to each voxel in the first analysis and to consciousness-related areas [39] thereafter. Theses areas will involve:The anterior insular cortex (AIC) [40]. Activation of the AIC has been associated with consciousness-related behaviors (subjective filling, attention, cognitive choices, intentions, music and time perception, awareness of sensation and movements… [39]);The posterior medial complex (precuneus plus adjacent posterior cingulate cortex) [41];The ascending reticular activating system [42];The thalamus [43,44] and its contribution to the anterior forebrain mesocircuit [45,46];The amygdala in the neural circuitry critical for emotion [47].

The association between cortical map activation and CRS-R will be studied for each patient and for each nosological group (especially for the MCS- and VS patient groups). If a specific pattern of cerebral activation can be highlighted for each nosological group in the early phase of coma, fPET neuroimaging will find a place in coma diagnosis, prognosis and therapeutic choices.

## Figures and Tables

**Figure 1 diagnostics-13-02026-f001:**
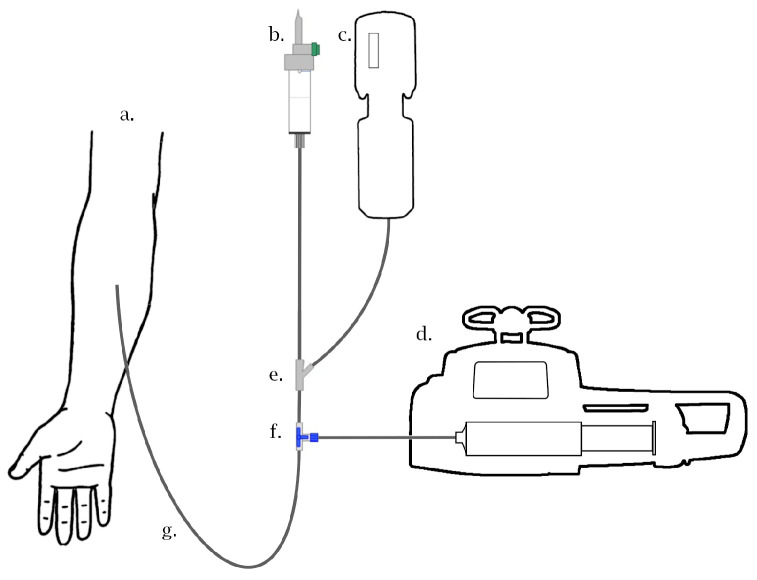
Infusion line scheme: (**a**) subject arm, (**b**) saline solution, (**c**) automated injector, (**d**) modified syringe pump, (**e**) y-connector with non-return valve to saline solution, (**f**) motorized 3-way stopcock, (**g**) syringe pump line.

**Figure 2 diagnostics-13-02026-f002:**
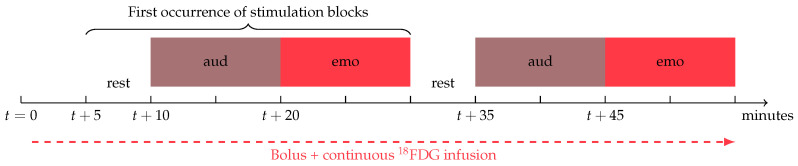
Stimulations during fPET. rest: no stimulation, aud: auditive stimulation, emo: emotional auditory stimulation and auditive stimulation.

**Figure 3 diagnostics-13-02026-f003:**
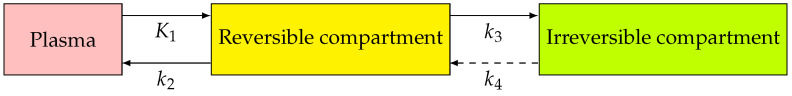
Two-tissue compartmental model. Dotted line shows the reverse rate constant (neglected) and solid lines show transport of ^18^FDG between different compartments.

**Table 1 diagnostics-13-02026-t001:** fPET protocol.

Timeline	Step
t−15 min	**Prepare infusion line** Install a 50 mL syringe in the infusion pump and purge the entire infusion line with saline solution. See Figure 1 for an illustration of the infusion line.
t−5 min	**Prepare ^18^FDG dose** Use the automatic radiopharmaceutical-dispensing system (KARL 100, Tema Sinergie) to prepare 2.8 MBq· kg${}^{-1}$ of ^18^FDG in automated injector (RadInject, Tema Sinergie): 25.5 kBq·kg${}^{-1}$·frame${}^{-1}$ [22]. Bring the automated injector into the PET room and connect it to the purged remote infusion system.
t−3 min	**Install patient** Install patient on the PET couch and connect infusion line to his/her catheter. Take the operator out of the PET room.
t−2 min	**Operate infusion system prototype** Remotely operate the 3-way stopcock (connection between automated injector and syringe). Remotely operate automated injector to inject tracer and continuously rinse the infusion line with saline solution. Filling sensor will close the 3-way stopcock when syringe reaches 50 mL.
t−1 min	**Perform scout and CT acquisition**
t = 0 min	**Start PET acquisition**
t + 10 s	**Make a 20% activity bolus followed by continuous infusion** Remotely operate 3-way stopcock to have connection between syringe and patient. Remotely operate modified syringe pump to make a 10 mL bolus. Remotely operate modified syringe pump to infuse continuous 40 mL at the rate of 48 mL per hour.
t + 5 min	**First occurrence of stimulation blocks** After reaching equilibrium of ^18^FDG arterial concentration, three task blocks will be repeated twice. First occurence: Five minutes of resting state (no sensory stimulation) Ten minutes of listening to a word list in an unknown language (Iranian word list) Ten minutes of listening to a close relative telling a story with affective speech.
t + 30 min	**Second occurrence of stimulation blocks** *Identical to first occurrence*
t + 55 min	**End of acquisition**

## Data Availability

We generate no clinical data in this protocol.

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
