# Peer review of "Functional PET Neuroimaging in Consciousness Evaluation: Study Protocol"

_diagnostics, 2023, doi:10.3390/diagnostics13122026_

Round 1

Reviewer 1 Report

Dr. T. Paunet et al. propose the protocol regarding the application of 18F-FDG-bolus plus infusion paradigm for monitoring the consciousness in coma patients using PET. This provides interesting insights, however there are several comments on this manuscript as follows.

1)    Throughout the manuscript, the expression of PET probe should provide a uniform style as 2-[18F]fluorodeoxyglucose or 2-18F-fluorodeoxyglucose and [18F]FDG or 18F-FDG, each of which “18” should be superscript character.

2)    Throughout the manuscript, the expressions of “glucose metabolism”, “glucose uptake”, “FDG uptake” are mixed for the “CMRGlu”. They may provide a uniform expression.

3)    In Abstract and 2.5. PET data acquisition, why is the ratio of FDG for bolus and infusion divided into 20% and 80%? Are there any evidences to support this ratio to be proper for quantitative assessment of CMRGlu?

4)    In Abstract and 2.2.1. Bolus plus infusion paradigm, how do you define as “emotional” auditory stimulation given by each legal representative. The degrees of emotional feeling may be difference between patients, which should affect the activation degree of CMRGlu..

5)    In Abstract, it is unclear what the situation of “after reaching baseline activity” means. Does it mean “after reaching equilibrium of FDG arterial concentration” as shown in Table 1?

6)    The similar question arises from 2.2.1. Bolus plus infusion paradigm. Authors describe “Task blocks begin 5 minutes after infusion start to reach a baseline state for vascular activity.”, however the time point of 5 minutes after infusion start can be obtain only once during a set of bolus plus infusion, meaning impossible to conduct multiple repeated tasks.

7)    In addition, how can the equilibrium of FDG arterial concentration be monitored? It seems no description regarding the method for arterial blood sampling.

8)    In Abstract and 2.4. Follow-up, why is the final follow-up period set at 18 months, not 12 or 24 months, post coma onset?

9)    In Table 1, 50-mL syringe is prepared, and after the 25 mL bolus, you plan to follow 40 mL (40 mL/hour) infusion, which means totally 65 mL needed. It seems to be 10 mL (20 % of 50 mL), not 25 mL for bolus injection.

10) Since authors try to apply their hand-made pump system, it should be strongly recommended to start from the test subjecting normal volunteers.

Reviewer 2 Report

The authors proposed a methodology belonging in an interesting area of research having great importance in every day clinical practice. However, due to its nature, the study that is realized and the methodologies that are proposed should be designed and carried out utilizing a complete and accurate setup. The identification of the parameters and the factors that influences that performance of the detection / characterization of the findings should be checked in depth minimizing the systematic errors (e.g. utilization of sample in performance validation of the system).

Moreover, some comments and points which should be under consideration are:

The authors should assign the advantages of their technique in comparison with the state of the art methodologies utilizing nowadays aiming at the same scope. A more recent literature review should be presented.

Sample’s dataset - real data input should be included in author’s database for the simulation of real situations in clinical environment. This might be an critical contribution to proposed study since in such a case the overall performance as well as the drawbacks of the proposed technique would be improved.

Round 2

Reviewer 1 Report

As a revised manuscript, the authors sincerely replied to all reviewers’ comments one-by-one.

Author Response

.